# Short Implants versus Longer Implants with Sinus Floor Elevation: A Systemic Review and Meta-Analysis of Randomized Controlled Trials with a Post-Loading Follow-Up Duration of 5 Years

**DOI:** 10.3390/ma15134722

**Published:** 2022-07-05

**Authors:** Miaozhen Wang, Feng Liu, Christian Ulm, Huidan Shen, Xiaohui Rausch-Fan

**Affiliations:** 1National Engineering Laboratory for Digital and Material Technology of Stomatology, Beijing Key Laboratory of Digital Stomatology, First Clinical Division, Peking University School and Hospital of Stomatology, National Clinical Research Center for Oral Diseases, 37A Xishiku Street, Xicheng District, Beijing 100034, China; wmiaozhen@126.com (M.W.); dentistliufeng@126.com (F.L.); shen_hd@pku.edu.cn (H.S.); 2Division of Oral Surgery and Implantology, Dental School, Medical University of Vienna, 1090 Vienna, Austria; christian.ulm@meduniwien.ac.at; 3Center of Clinical Research, Division of Conservative Dentistry and Periodontology, Dental School, Medical University of Vienna, 1090 Vienna, Austria

**Keywords:** dental implant, short implants, vertical augmentation, systematic review, failure rate, meta-analysis

## Abstract

This study compared the outcome of fixed prostheses supported by short implants (<8 mm) and longer implants (≥8 mm) with an elevated sinus floor after 5 years of follow-up. The literature searches were performed electronically and manually in PubMed, EMBASE, and Web of Science databases to identify relevant articles published from 1 January 2013 to 31 January 2020. We selected eligible studies using inclusion criteria and assessed their quality. From 1688 identified studies, five randomized controlled trials were included. Between the short implant group and the control group, the implant failure-related pooled risk ratio (RR) was 3.64 (*p* = 0.07). The RR for technical complications was 2.61 (*p* = 0.0002), favoring longer implants. Marginal peri-implant bone loss after 1 and 5 years of function showed statistically significant less bone loss at short implants (1 year: mean difference = 0.21 mm; *p* < 0.00001; 5 years: mean difference = 0.26 mm; *p* = 0.02). The implant failure and the biological failure of both groups were similar after 5 years of follow-up. Short implants could be an alternative to long implants with an elevated sinus floor for atrophic maxillae in aging populations. Studies with larger trials and longer periods of follow-up (10 years) remain essential.

## 1. Introduction

Over the last 10 years, short implants in the posterior jaw have been used extensively to minimize complicated surgical procedures, e.g., sinus floor elevation and distraction osteogenesis [1]. Recent reviews suggested that short implants were a preferable alternative treatment because of their lower cost, surgical time, and complication rates [2,3,4]. However, short implants in the posterior maxillae have shown variable success rates. A retrospective study of 4591 implants reported that short (6 mm) implants in the maxillae have a significantly lower survival rate (87%) than those in the mandible (100%) after up to 10 years of follow-up [5]. This difference might have been caused by insufficient posterior maxillae bone quantity and quality. However, other studies demonstrated that short implants (5–7 mm) have success rates ranging from 90% to 98% in the posterior maxillae after 5–10 years of follow-up [1,6,7].

There is controversy surrounding the definition of a short implant, with some authors considering “short” to be 7–10 mm [8], whereas other authors define “short” as those implants with a design intra-bone length of 8 mm or less [9]. The fourth European Association for Osseointegration (EAO) consensus conference [10] reported that a short implant (<8 mm) had a survival rate of 99.0% after 16–18 months of follow-up in routine treatment. Therefore, we undertook a systematic review and meta-analysis in which “short” described implants of less than 8 mm according to the definition proposed by Plonka et al. [11].

Over recent years, a few systematic reviews [12,13,14,15] have assessed the efficacy of short implants compared with longer implants combined with sinus floor elevation in atrophic posterior maxillae; in most of them, the follow-up duration of the included randomized controlled trials (RCTs) trials were too short (1–3 years) to draw reliable conclusions. Concerning the limitation about implant length and loading time in the literature research mentioned above, it is necessary to perform a systematic review of RCTs focusing on the outcomes of maxillary prostheses supported by implants that are shorter than 8 mm compared with similar prostheses supported by longer implants (≥8 mm) combined with an elevated sinus floor with follow-up of 5 years of function. A longer follow-up is desirable, but it is not currently available.

Therefore, the aim of the present study was to evaluate if short implants placed in the posterior atrophic maxilla can be as effective as longer implants associated with an elevated sinus floor, assessing factors such as implant survival, marginal bone loss, and complications after at least 5 years of function.

## 2. Materials and Methods

This study was carried out following the PRISMA statement guidelines [16]. The present study focused on whether the implant failure rate and other clinical outcomes differed between short and long implants supporting fixed prostheses in RCTs.

### 2.1. Eligibility Criteria

The eligibility criteria for studies comprised:RCTs: patients with maxillary atrophy, with 4–7 mm of bone height, were randomized into two groups. One group was rehabilitated with prostheses supported by short implants (<8 mm) without bone grafts. The other was rehabilitated by longer implants combined with sinus floor elevation;The study was randomized and included the use of short (<8 mm) and longer implant groups; all patients (male or female) were ≥18 years old;The implants were evaluated with an average follow-up duration of at least 5 years after loading;Each group included more than 10 patients.

The primary outcomes were: implant failure and biological and technical complications in the short and longer implant groups.

In addition, the difference in peri-implant marginal bone levels (MBLs) between short implants and longer implants in the posterior maxilla at 5 years of follow-up was considered the secondary outcome.

The exclusion criteria were as follows: all retrospective studies, in-vitro studies, animal studies, professional opinions, review articles, case reports, case series, cohort studies, and studies that did not report the above-mentioned primary and secondary outcomes.

### 2.2. Search Strategy and Information Sources

Databases including PubMed, EMBASE, and Web of Science were searched electronically to identify all relevant articles published from 1 January 2013 to 31 December 2019. The search was limited to the English language, and this meta-analysis used the keywords: “dental implant”, “short dental implant”, “extra short implant”, “ultra-short implant”, “short implant” AND (“Maxilla” OR “maxillae” OR “jaws” OR “Dental Arches” OR “partially edentulous”) AND (“sinus” OR ”crestal approach” OR ”lateral approach” OR ”trans-crestal sinus floor elevation”). OR was used to combine these terms to carry out the search.

In addition, searching by hand was carried out from January 2013 to December 2019 in dental implant-related journals, including *Clinical Implant Dentistry and Related Research*, *Clinical Oral Implants Research*, and the *European Journal of Oral Implantology*.

Manual searching of the reference lists of previously published systematic reviews, meta-analyses, and relevant papers was carried out.

### 2.3. Study Selection

The citations (titles and abstracts) were screened independently by two investigators (M.W. and F.L). The same investigators then evaluated the complete text of the eligible studies, selecting those articles that met the inclusion criteria. Any disagreement on the title, abstract, or full text was reassessed by a third author (C.U.). Articles that did not match the inclusion criteria and duplicated articles were excluded from further review. If more than one publication used the same patient population, only the study with the longest follow-up period was included.

### 2.4. Risk of Bias

The Cochrane collaboration tool [17] was used to assess the risk of bias in all the included RCTs. The quality of the methods used in the included studies was evaluated independently by two authors. Questions were defined as related to the risk of bias and were answered as yes (low bias risk), no (high bias risk), or unclear. The answers were scored as no and unclear = 0 and yes = 1, and the scores of eight items were totaled to generate the total quality score of each article. A third author was involved in resolving any disagreements by consensus. The quality of the methods was assessed as high ≥ 6, moderate = 5, and weak ≤ 4. A funnel plot was used to assess the risk of publication bias across studies.

### 2.5. Data Extraction

A data extraction form was developed to record the data extracted by two investigators (M.W. and F.L.) independently, and data were presented from each included study. Disagreements were resolved by consensus or with input from a third author (X.R.-F.).

### 2.6. Synthesis and Meta-Analysis of the Data

Primary outcomes comprised implant failure and biological and technical complications in the short and longer implant groups.

Implant failure: implants that were not present in the mouth or that did not match any of the success criteria were judged to have failed;Biological complications (intra- or postoperative): sinus membrane perforations, sinusitis (acute sinus infection), soft tissue dehiscence, fistula, swelling, infection, or implant failure;Technical complications: abutment screw loosening or screw fracture, crown chipping or loss retention, replacing the crown;Peri-implant MBL: the distance between the implant shoulder and most coronal point of bone-to-implant contact at each evaluation time point (implant placement at baseline and the follow-up times).

The primary outcomes were evaluated using the risk ratio (RR). Forest plots of MBL, as a continuous outcome, were constructed using weighted mean differences. Meta-analyses were carried out using the statistical software Review Manager (Version 5.3. The Nordic Cochrane Centre, Copenhagen, Denmark).

The statistical unit for the implant failure outcomes was referred to the implant. The statistical unit for the other outcomes was referred to the patient. The I^2^ statistic was employed to represent the percentage of the total variation across studies caused by heterogeneity, in which 0–25% corresponded to low heterogeneity, 25–50% corresponded to moderate heterogeneity, and 50–100% represented high heterogeneity. For fixed-effects or random-effects models, we applied the inverse variance method. If heterogeneity was detected as statistically significant (*p* < 0.10), we used a random-effects model to assess the significance of treatment effects. In the absence of such heterogeneity, the data were analyzed using a fixed-effects model. The relative effect for dichotomous outcomes was estimated and shown as the risk ratio (RR) and the mean difference (MD) in bone thickness is expressed in millimeters for continuous outcomes, both with their associated 95% confidence interval (CI). Statistical significance was considered at *p* < 0.05. Meta-analysis was attempted only for studies with similar comparisons reporting the same outcome measures. In cases where no events (or all events) were observed in either group, the study was judged to yield no information regarding the relative probability of the event, and such studies were omitted automatically from the meta-analysis. In these cases, the term “not estimated” is shown in the RR column of RR of the table.

## 3. Results

### 3.1. Literature Search

The initial literature search identified 1688 articles. After the screening of the abstracts and titles, 24 possibly pertinent articles remained. Examination of the obtained full-text articles revealed that only five studies satisfied the inclusion and exclusion criteria. Figure 1 shows the PRISMA-based flowchart of the selection process.

### 3.2. Description of the Studies

The excluded studies, together with explanations for their exclusion, are listed in Table 1.

Table 2 shows information regarding the five selected RCTs. Two of the RCTs had a split-mouth design, and three had a parallel-group design. According to these articles, 406 implants (197 short implants and 209 longer implants) were installed in 217 patients whose average age ranged from 48 to 61 years.

All the studies in this review excluded patients who had received radiation and chemotherapy, those treated with an intravenous bisphosphonate, and those with uncontrolled diabetes [18,19,20,21,22].

The length of the short implants ranged from 5 to 6 mm. Among the 197 short implants, 70 were 5 mm long (36%), and 127 were 6 mm long (64%). In terms of implant connection types, 122 short implants were connected internally (122/197, 61.9%), and 36 short implants were connected externally (75/197, 38.1%). Among the 209 control implants (length = 10–15 mm), 128 (61.2%) had an internal connection, and 81 (38.8%) had an external connection.

In Pietro’s study [19], longer implants were placed 4 months after sinus floor elevation, whereas in the other four studies, placements of longer implants using single-stage procedures were used [18,20,21,22]. In these studies, there was a healing time of 3–5 months between implant placement and provisional loading.

### 3.3. Assessment of Study Quality

The Cochrane Collaboration tool [17] was used to assess the five selected RCTs. Table 3 shows the analysis of the bias risk of all the RCTs. All five studies reported appropriate randomization of the subjects, and the results were reported clearly.

Blinding of surgeons and patients was difficult because the patients must be informed concerning the type of implants used for treatment, and the surgeons would know which implant type had been used during surgery. Consequently, all five studies were considered to be at high risk of bias.

In the selected five studies, the assessment of clinical outcome could not be blinded completely because, during the radiographic assessment, independent investigators could observe the specific shape and length of implants [18,19,20,21,22].

Group imbalance-related bias. Pietro et al. [19] employed implants of various diameters in the two groups: generally, the diameters of the short implants were wider (6 mm) than those of the control implants (4 mm or 6 mm). The other three studies used short implants and control implants with the same diameters. In Esposito’s study, the diameters for both groups were 5 mm [20]. In the other three studies, the diameters for both groups were 4 mm [18,21,22]. Moreover, in two of the studies, the prostheses had a splint-crown design [19,20], while the other studies used single crowns [18,21]. These factors meant that there was a high risk of outcome bias.

Sample size-associated bias. A study was evaluated as low risk if it recruited sufficient participants to have 80% power. However, the power calculation was usually only conducted for the primary outcome, e.g., bone level change, which might have been different from the outcome analyzed in this review.

The duration of follow-up. The study with more than 5 years of follow-up [23] was assessed as having a low risk of bias, as were the other four studies.

Conflict of interest-related bias. All five included studies mentioned their sponsors but claimed sole ownership of the data and stated that the sponsors did not interfere with the trial conduct or its publication.

Radiographic outcome-related bias. The radiographic assessment could not be carried out in a blinded manner, meaning that all five studies were adjudged as having a high risk of radiographic outcome-related bias.

Clinician-related Bias. In three studies [19,20,22], all treatments were carried out by the same surgeon and prosthodontist; however, two studies [18,21] did not clearly address which clinicians performed the treatment.

### 3.4. Implant Failure

In the five selected studies, 8 out of 406 dental implants failed (2.0%), comprising seven short implants (3.5%) and one longer implant (0.5%) during 5 years of follow-up.

The results for the two groups were compared using a meta-analysis. In the comparison of implant failure between short implants and longer implants, the pooled risk ratio (RR) was 3.64 (95% confidence interval (CI): 0.91–14.53) (Figure 2), implying that long implants are more favorable. However, the failure rate between the two groups was not statistically significantly different (*p* = 0.07).

The failure rates between external connection implants and internal connection implants were similar (4% vs. 5.7%) for the short implant group (Table 2). In the longer implant group, only one implant with an internal connection failed.

### 3.5. Complications

The meta-analysis of the five trials for biological complications did not show any statistically significant difference between the two groups (RR = 0.95; 95% CI: 0.51–1.77; *p* = 0.86; chi-squared (Chi^2^) = 7.27, df = 4 (*p* = 0.12); I^2^ = 45%) (Figure 3).

For technical complications between the short implants and the control implant, the RR was 2.61 (95% CI: 1.57–4.34; *p* = 0.0002; Chi^2^ = 3.73, df = 4; I^2^ = 0%), significantly favoring longer implants (*p* = 0.003) (Figure 4). A list of the complications reported from all the trials is presented in Table 4.

### 3.6. Changes in Peri-Implant Marginal Bone Levels

All the included studies noted changes in the marginal bone level (MBL). Meta-analysis of the five trials for marginal peri-implant bone loss after 1 year of function (Figure 5) showed statistically significant less bone loss using short implants (mean difference = 0.21 mm; 95% CI: 0.37–0.05; *p* = 0.009; Chi^2^ = 38.93, df = 3 (*p* < 0.00001); I^2^ = 92%). Meta-analysis of the five trials for marginal peri-implant bone loss after 5 year of function (Figure 6) also showed statistically significant less bone loss using short implants (mean difference = 0.26 mm; 95% CI: 0.47 0.05; *p* = 0.02; Chi^2^ = 7.64, df = 4 (*p* = 0.11); I^2^ = 48%).

**Table 1 materials-15-04722-t001:** Description of excluded studies.

Reason	References
Data from the same group of samples	Pietro et al. [24]
	Pietro et al. [25]
	Roberto et al. [26]
	Pohl V et al. [7]
	Schincaglia et al. [27]
	Marco et al. [28]
	Felix et al. [29]
Follow-up time < 5 years	Marco et al. [30]
	Sahrmann et al. [31]
	Pietro et al. [32]
	Gastaldi et al. [33]
	Bolle et al. [34]
	Jun-Yu et al. [35]
	Taschieri et al. [36]
	Bechara et al. [37]
	Hadzik et al. [38]
	Shah et al. [39]
	Amir et al. [40]
Length of short implant not less than 8 mm	Cannizzaro et al. [41]

**Table 2 materials-15-04722-t002:** Characteristics of the included studies.

Study	Unicenter/Multicenter	Total No. of Patients/Implants	No. of Implants in Each Group		Patients’ Age Range (Average) (Years)	Definitive Restoration	Brand and Connection	Dimension of the Implant (Length × Diameter)
Felix [18]	Multicenter	41/41	Short	21	50 (30–71)	Single crown	Astra, internal	6 × 4
			control	20	48 (29–72)			11 × 4
Pietro [19]	1	15/72	Short	34	56 (45–70)	Splint crowns	Megagen, internal	5 × 6
			control	38				10 × 6 or (10,11.5,13) × 4
Marco [20]	Multicenter	20/83 41/41	Short	36	61.1 (45–70)	Splint crowns	Megagen, external	5 × 5
			control	37	58.5 (45–75)			(10,11.5,13,15) × 5
Thoma [21]	Multicenter	15/72 40/73	Short	67	50 (23–76)	single	Astra, internal	6 × 4
			control	70	51 (20–77)			(11,13,15) × 4
Pietro [22]	2	101/137 20/83	Short	39	57.6 (45–80)	Single/ splint crowns	Southern, external	6 × 4
			control	44				(10,11.5,13,15) × 4

**Table 3 materials-15-04722-t003:** Quality assessment of included studies.

Bias	Felix (2019) [18]	Pietro (2019) [19]	Marco (2019) [20]	Thomma (2018) [21]	Pietro (2019) [22]
Random sequence generation					
Allocation concealment					
Blinding of patients and surgeons					
Blinding of outcome assessment					
Incomplete outcome data					
Selective reporting					
Other sources of bias
Group imbalance					
Sample size			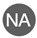		
Follow-up time					
Conflict of interest					
Radiographic outcome					
Clinician bias					
Score	7	8	7	7	8

**Table 4 materials-15-04722-t004:** List of complications by study group.

Study	No. Implants in Each Group		Number of Patients with Complications (Number of Complications)		In Total
			Biological Complications	Technical Complications	
Felix [18]	Short	21	1 implant failure	3 replaced crowns, 1 chips, 1 abutment loosened	4p (7) (n = 19)
	Control	20	0	1 abutment loosened	1p (1) (n = 17)
Pietro [19]	short	34	3 perforation, 1 mucositis, 3 implant failures	Prosthesis decementation 1	5p (5) (n = 15)
	control	38	1 sinus perforation, 1 implant failure	prosthesis decementation 1	2p (2) (n = 15)
Marco [20]	short	36	1	1 replaced crown, 1 failure due to implant	2p (2) (n = 17)
	control	37	5p membrane perforated	0	7p (7) (n = 19)
Thoma [21]	short	67	Fistula, swelling, infection or implant failure: 5	Abutment loosened, fracture, chipping, lost crown or loss retention: 21 (47.7%)	26 (n = 40)
	control	70	Fistula, swell, infection or implant failure: 9	Abutment loosened, fracture, chipping, lost crown or loss retention: 14(30.4%)	23 (n = 53)
Pietro [22]	short	39	Implant failure	0	1p (n = 17)
	control	44	4 perforations	1 chip	5p (n = 17)

## 4. Discussion

In this review, only RCTs were included, which, compared with not RCTs, yielded a higher level of evidence and enhanced the strength of the results. We included five RCTs [18,19,20,21,22] that compared fixed prostheses supported by short implants (<8 mm) with those supported using longer implants (≥8 mm) combined with elevation of the sinus floor in the atrophic maxilla, with 5 years of functional follow-up. This review revealed no difference between the two groups concerning biological failure. Similarly, for implant failure, there were no clinically significant differences between short and longer-length implants; however, longer implants were preferred. Significantly less marginal bone loss was found in the short implant group after 1 and 5 years of follow-up. However, short implants had more technical complications compared with longer ones.

### 4.1. Definition of a Short Implant

There is controversy surrounding the definition of a short implant, with some authors considering implants of 6 to 10 mm in length to be “short” [8,42,43], whereas other authors define “short” as those implants with a design intra-bone length of 8 mm or less [9]. At present, implants as short as 4 mm long are available for clinical use. As alternatives to bone augmentation procedures, short implants were demonstrated to be predictable and reliable [4,44,45], in addition to their generally accepted advantages such as reduced treatment costs, postoperative morbidity, and intra-operative time. The clinical efficacy of short dental implants in comparison with longer implants alone [4,43,46,47] or combined with staged or simultaneous maxillary or vertical bone augmentation [10,13,14] has been the subject of recent systematic reviews and meta-analyses. The majority of these reports revealed survival rates in excess of 95% using short dental implants compared with longer implants during 1 to 3 years of follow-up, although most previous studies considered 8 mm or less as a short implant. Several recent clinical studies confirmed that 8 mm implants gave predictable results in the mandible and posterior maxilla and have been used widely, which could be considered standard implants [11,43]. Thus, in the present systematic review and meta-analysis, “short implant” was defined as implants with an intra-bone length less than 8 mm, and implants with a length of 8mm or more were “longer implants” [11].

### 4.2. Implant Failure

None of the included RCTs reported a statistically significant difference in implant failure between the control and test groups [18,19,20,21,22], which was confirmed by the results of the meta-analysis of those trials. Recent meta-analyses and systematic reviews supported the lack of a significant difference between short implants and standard implants in terms of implant survival [4,12,14,23,44,48,49]. When considering the biomechanical disadvantages of short implants, more failures might be expected compared with long implants after loading. Recent long-term studies have shown that the risk of short implant failure increases significantly from the third to the fifth year during function [1,50,51]. However, our analysis of the results of RCTs showed no increase in the risk of failure for short implants after 5 years of function, which should be interpreted with caution because the data tended to favor longer implants (RR = 3.64). A potential reason causing the difference may be the different subjects we focused on. These three studies [1,50,51] focused on sites with sufficient bone volume. That means longer implants were placed in native bone. On the contrary, our research subjects were the sites where the bone height was insufficient; thus, longer implants were placed in grafted bone. However, it seems reasonable to assume that continuing or rapid loss of peri-implant bone around short implants could be more critical compared to longer implants, and it might cause a higher rate of loss of implants. Therefore, it is crucial to control the main risk factors for peri-implant diseases and to apply strict maintenance programs for the long-term performance of these implants.

According to Pietro’s study, three short implants failed [19]. One implant was successfully replaced by one that was placed more distally and loaded. The other two failed implants were not replaced because the patient did not want to rehabilitate the area for a second time. Esposito et al. found one short implant failed 3 months after loading with its provisional crown [20]. It was successfully replaced by a wider diameter rescue implant. The other three articles did not show enough details about the failed implants [18,21,22]. Due to the small trauma and short treatment time of short implants, low rate of failure, and the possibility of replacing a new implant if it fails, short implants might be a choice for the rehabilitation of posterior atrophy maxillary in old patients.

### 4.3. Complications

The biological complications during the entire study period included fistula, swelling, infection, and implant failure. The meta-analysis of the five RCTs showed 35 overall biological complications, comprising 15 associated with short implants versus 20 associated with longer implants; however, there were no statistically significant differences between the two groups.

Technical complications included fracture of the abutment screw, screw loosening, chipping of veneering ceramic, lost crowns, and loss of retention and replacing crowns. Our results showed that shorter implants have additional technical complications after 5 years of loading. This might be because, compared with longer (control) implants, shorter implants might experience detrimental loading forces. Our results support previous findings that the risk of short implant failure increases significantly from 3 to 5 years of function [1,50,51]; thus, technical complications should be considered major risk factors.

### 4.4. Marginal Bone Loss

Our results are encouraging; apparently, short implants lost 0.21 mm less peri-implant marginal bone than longer implants at 1 year after loading and 0.26 mm less after 5 years of loading. The result that short implants suffer less marginal bone loss than longer implants was also observed in several clinical trials during 3 years of follow-up [4,12,44,48,49,52]. It was claimed that a detrimental crown-to-implant ratio could theoretically cause biomechanical issues and excessive stress on the marginal bone; however, there was a lower change in MBL around short implants compared with longer ones during 5 years of follow-up. The potential hypotheses required further research.

Stefan et al. inferred that the marginal bone loss was a result of natural changes in the bone’s morphology in response to the implant, such as a decrease in the convexity of the outer surface and concavity of the inner bone [53]. They found that when implants showed a vertical bone loss, apical bone part rather increased. Hence they questioned the demand for “large implant surfaces and lengths” as it is traditionally set up in conventional dental implantology.

### 4.5. Placement of Short Implants in Maxillary Sinus Augmented Sites

In our research, short implants presented similar clinical results to longer implants when combined with elevation of the lateral sinus floor. However, caution should be exercised when short implants are used in cases of an augmented maxillary sinus [54,55]. Deporter et al. reported a loss rate of 14% for short implants in the augmented maxillary sinus [54], while Sohn et al. reported a loss rate of 9% [55]. In these two studies, the 106 short implants used comprised 20 that supported single crowns and 86 that supported splinted crowns. The meta-analysis demonstrated a slight preference for single crowns in Deporter et al.’s study, whereas splinted crowns were preferred in Sohn et al.’s study.

### 4.6. Limitations

It is worth mentioning that several factors might have affected the results, such as the design of the implant, occlusal loading, crown splinting, and the crown height space, which in this systematic review are considered limitations [4,23,48]. First, the present review combined different types of prosthetics for analysis, which caused difficulties in results interpretation and in making strong conclusions. The presence of a detrimentally high crown/implant ratio results in high occlusal forces, which have been suggested to be distributed better using multiple splinted short implants [56,57]. Consistent with the above suggestion, a systematic review compared 6 mm implants to standard implants for single-crown restorations in the posterior region, demonstrating that during 3 years of follow-up, the survival rate of short implants was poorer than that of standard implants [45].

Combining the data from RCTs [18,19,20,21,22] and systematic reviews [12,14,44], as well as the present findings, suggested that short implants after at least up to 5 years of loading might be a feasible alternative to standard implants in combination with sinus floor elevation. However, the long-term prognosis of short implants is undetermined, and the small sample sizes of the present and previously published RCTs do not allow definitive conclusions to be drawn. Hence, more RCTs with larger sample sizes and 10 years of follow-up are required. In particular, the possible impact of different types of prosthetics using a single crown or splint crown implants should be investigated. More high-quality studies with large data sets for hierarchical analysis are required in the future. When considering aspects of long-term stability and safety, the current opinion still suggests that short implants are more available for clinical use in an aging population.

## 5. Conclusions

Over 5 years of follow-up, there were no significant differences in implant failure or biological failure between the group receiving short implants and the group receiving longer implants.

Significant less marginal bone loss was found in the short implant group during follow-up for 1 and 5 years. Therefore, short implants could be an alternative choice to long implants with sinus floor elevation in the atrophied maxilla of the aging population. However, it should be emphasized that when placing a short implant in an atrophied maxilla, more technical complications might be expected after long-term follow-up. Larger trials with a longer period of follow-up (10 years) remain essential to compare the long-term outcomes of both procedures to help clinicians make optimum therapeutic decisions for their patients individually.

## Figures and Tables

**Figure 1 materials-15-04722-f001:**
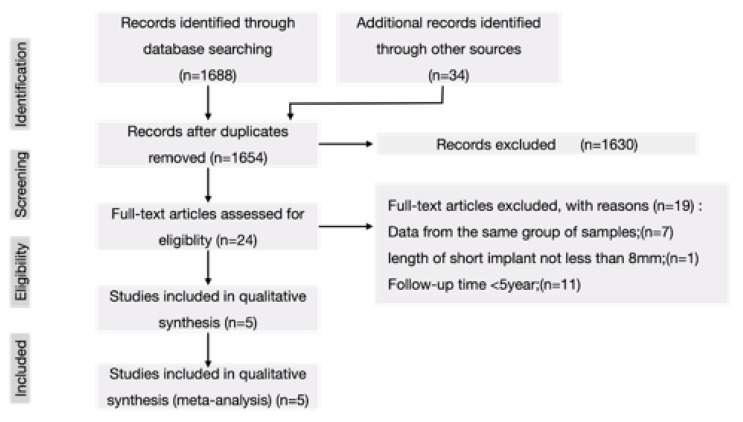
The search strategy of the present study according to PRISMA guidelines.

**Figure 2 materials-15-04722-f002:**
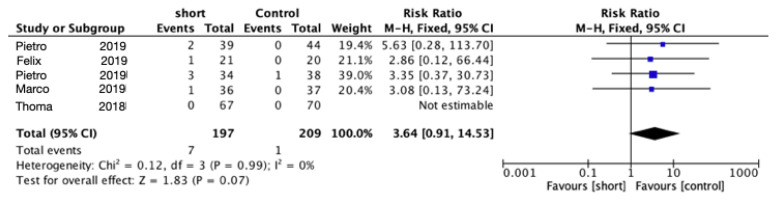
Comparison of implant failures using short implants vs. longer implants with elevation of the sinus floor after 5 years in function via a meta-analysis.

**Figure 3 materials-15-04722-f003:**
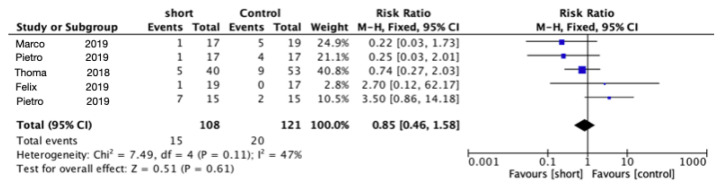
Comparison of biological complications using short implants vs. longer implants with elevation of the sinus floor after 5 years in function via a meta-analysis.

**Figure 4 materials-15-04722-f004:**
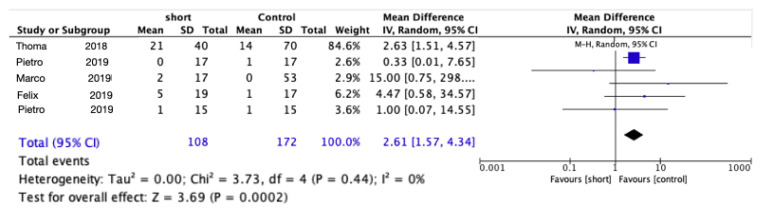
Comparison of prosthetic complications using short implants vs. longer implants with elevation of the sinus floor after 5 years in function via a meta-analysis.

**Figure 5 materials-15-04722-f005:**
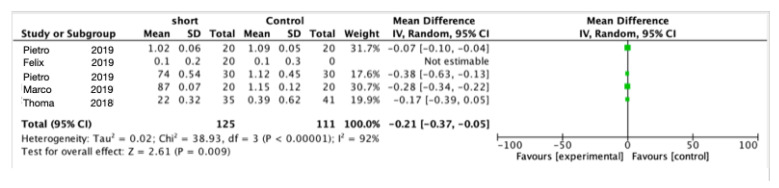
Comparison of marginal bone loss using short implants vs. longer implants with elevation of the sinus floor after 1 year in function via a meta-analysis.

**Figure 6 materials-15-04722-f006:**
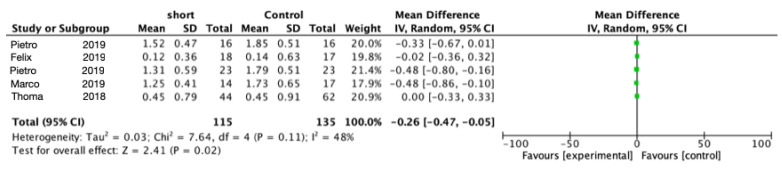
Comparison of marginal bone loss using short implants vs. longer implants with elevation of the sinus floor after 5 years in function via a meta-analysis.

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
