# Peer review of "Short Implants versus Longer Implants with Sinus Floor Elevation: A Systemic Review and Meta-Analysis of Randomized Controlled Trials with a Post-Loading Follow-Up Duration of 5 Years"

_materials, 2022, doi:10.3390/ma15134722_

Round 1
Reviewer 1 Report
Thnak you for the oportunity to review the article entitled "Short implants versus longer implants with sinus floor elevation: A systemic review and meta-analysis of randomized controlled trials with a 5-year post-loading follow-up"
In my opinion, ther paper is not properly written when it comes to structure, it's design, presentation and execution . The tables are hard to read and understand especially tabl. 3. A systematic review based on 5 RCT is not good enough to draw scientific value conclusions especially as the authors noticed in limitation section with "different types of prosthetics" . Thus it should be not granted for publication in a high impact-factor journal. It simply does not meet the standards of scientific soundness and its overall merit is low.
Author Response
On behalf of our coauthors, we thank you for allowing us to revise our manuscript. We appreciate the suggestions and instructive comments on our manuscript, which helped us to greatly improve its quality. The tables have been revised for easy understanding.
Although only 5 RCT studies were included in this systematic review, these studies were high-quality randomized clinical studies that provide high-quality evidence. Prospective or retrospective cohort studies, single prospective or retrospective studies and other non-randomized controlled trials were excluded. Thus, we reduced the influence and risks to the final conclusion caused by low-quality articles due to bias in experimental design and reduced the total number of included cases. Considering that only 5 articles are included in this study, The sample size will be too small to draw meaningful conclusion considering the possible impact of different types of prosthetics if they are divided into single crown group and splint crown group. We looking forwards to included more high-quality studies with large data sets for hierarchical analysis in future. Thank you very much for your valuable recommend about the language. The manuscript has been professionally edited by a native English speaker.
Thank you for your time in reviewing our resubmission. We look forward to hearing from you.
Yours sincerely
Miaozhen Wang
Reviewer 2 Report
the bibliography is poorly referenced in the text
The bibliography goes from 7 to 18After 23 it returns to 8
Line 181 1-17.49 without courses and returns to 49
Line 236 says:
"The failure rates between externally connected implants and externally connected implants were similar (4% vs. 5.7%) for the short implant group. In longer implant group, only one implant with an internal connection failed." Repits 2 times externally
Author Response
On behalf of our coauthors, we thank you for allowing us to revise our manuscript. The bibliography has been rearranged in an ascending order.
Thank you for your valuable comments. The spelling mistakes have been corrected in line 236 according to your suggestion.
Thank you for your time in reviewing our resubmission. We look forward to hearing from you.
Yours sincerely
Miaozhen Wang
Reviewer 3 Report
This study was very excellent.
1) I would like to ask your answers regarding the risk of short implant failure increasing significantly from 3 to 5 years of loading.
2) when you mentioned aging population as a target group for short implants, have you consideration of the effects of repeated implant surgery due to failure of short implants? Please discuss this problems, too.
Author Response
Thank you for your constructive suggestion. We have added more discussion in the article as you suggested.
- About Short implant failure increased from 3 to 5 years of loading.(line 561-570)
A potential reason causing the difference may be the different subjects we focused. These three articles studied focused on sites with sufficient bone volume. That means longer implants were placed in native bone. On the contrary, our research subjects was the sites that the bone height was insufficient, thus longer implants were placed in grafted bone. However, it seems reasonable to assume that continuing or rapid loss of peri-implant bone around short implants could be more critical comparing to longer implants, and it might caused a higher rate of loss implants. Therefore, it is crucial to control the main risk factors for peri-implant diseases and to apply strict maintenance programs for the long-term performance of these implants.
(2) about effects of failure of short implants (line 571-601)
Accoridng to Pietro’s study, three short implants were failed[15]. One implant was successfully replaced by one that was placed more distally and loaded. The other 2 failed implants were not replaced because the patient did not want to rehabilitate the area at a second time. Esposito et al found one short implant failed 3 months after loading with its provisional crown[16]. It was successfully replaced by a wider diameter Rescue implant. The other 3 articles did not show enough details about the fail implants[14,17,49]. Due to the small trauma and short treatment time of short implants, low rate of failure, and the possibility to replace a new implant if it fails, short implants might be a choice for the old patients with an atrophy maxillary.
Thank you for your time in reviewing our resubmission. We are looking forward to hear from you.
Yours sincerely
Miaozhen Wang
Reviewer 4 Report
Reviewer: To the authors:
I was invited to review this manuscript.
Title: Short implants versus longer implants with sinus floor elevation: A systemic review and meta-analysis of randomized controlled trials with a post-loading follow-up duration of 5 years
Considering its content and presentation, in my opinion, this manuscript may have the merit to be accepted for publication after a minor review.
There are numerous typographical and grammatical errors throughout the paper.
Line 16 “ in PubMed” not only I suggest ”PubMed, EMBASE and Web of Science”
Lines 40-41 “…had success…” I suggest “…has success…” “after 5-10 years follow-up”
Lines 58-59 A longer follow-up is desirable, but it is not currently available
Line 62 “as the survival of the implant” I suggest “as implant survival”
Line 126 “We developed". I suggest “It was developed”
Line 127 “are” or “were”
Line 131 Title using italics
Line 176 “splint-mouth” I suggest “split-mouth”
Line 181 format references [14-17,49]
Line 188 “Pietro Felice’s”. Pietro is the name, Felice is the surname. Please check the manuscript and uniform formatting according to Guidelines for Authors
Line 194 the reference is not correct please check
Line 206 What studies?
Line 216 “greater than 5 years” is better “more than 5 years”
Line 232 “long” I suggest “longer”
Line 236 There is a error in the sentence. External or internal connection implants? And is there a figure for this data?
3.5 Complications: this section is in need of an accurate check. The datas are wrong. Please verify
Thera is a problem with names and surnames in all Figures and Tables (e.g. Felice P, Felix LG, Felice P, Esposito M,Thoma DS) Please check!
Line 289 Correct follow-up time < 5 years
Line 297 level of evidence
Line 298 RCTs and not RCTS
Line 302 “between shorter and longer length implants” I suggest “between short and longer implants”
4.1 Definition of a short implant: this section is in need of an accurate check. Unclear sentences
Lines 322 and 324 format [23,30] and [27,30]
Lines 328 “according with” I suggest “according to”
Lines 360-363 I would like the authors to explain better
Lines 393-396 I would like the authors to explain better
References is in need of a thorough revision. Please check formatting, according to Guidelines for Authors.
The only not so positive point is the limited mastery that the authors seem to have in the English language. I saw many mistakes and pointed some of them out, but the paper needs a full linguistic revision before being eligible for publishing.
Author Response
Thank you very much for your valuable comments and professional suggestions.
1)All typographical and grammatical errors you listed have been corrected according to your suggestions.
2)Data of the complication has been checked and corrected.
3)Names and surnames in Figures and tables have been corrected too.
4) definition of a short implant, this section has been checked and described in Line 531-549 according to your kindly suggestion.
5) we have checked the formatting of references as your kindly suggestion.
6) Lines 360-363 have turned to Line 618-622,We re-discuss this part as follows:
Our results are encouraging, apparently short implants lost 0.21 mm less peri‑implant marginal bone than longer implants at 1 year after loading, and 0.26 mm less after 5 years of loading. The result that short implants suffer less marginal bone loss then longer implants was also observed in several clinical trials during 3 years of follow-up. It was claimed that a detrimental crown-to-implant ratio could theoretically cause biomechanical issues and excessive stress on the marginal bone; however, there was a lower change in MBL around short implants compared with longer ones during 5 years of follow-up. The potential hypotheses required further research.
7)Lins 393-396 have turned to Line 667-672,We re-discuss this part as follows:
Hence, more RCTs with larger sample sizes and 10 years of follow-up are required. In particular, the possible impact of different types of prosthetics using single crown or splint crown implants should be investigated. More high-quality studies with large data sets for hierarchical analysis are required in future. Considering aspects of long-term stability and safety, the current opinion still suggests that short implants are more available for clinical use in an aging population.
Thank you for your time in reviewing our resubmission. We look forward to hear from you.
Yours sincerely
Miaozhen Wang
Round 2
Reviewer 1 Report
Thank you for the opportunity to review the corrected version of manuscript titled "Short implants(<8mm) versus longer implants with sinus floor elevation: A systematic review and meta-analysis of randomized controlled trials with a post-loading follow-up duration of 5 years.
The Authors made some corrections regarding tables and in the text in the discussion section in regards to MBL they may find interested to discuss and refer to recently published paper https://doi.org/10.3390/app12115589
Author Response
Thank you very much for your comments and kindly suggestion. We have added some discussion according to your advice as follows which was in line 626-631.
Stefan et al infered that the marginal bone loss was a result of nature changes of the bone’s morphology in response to implant such as a decrease in convexity of the outer surface and concavity of the inner bone[57]. They found that when implants showed vertical bone loss, apical bone part rather increased. Hence they questions the demond for “large implant surfaces and lengths” as it traditionally set up in conventinal dental implantology.
Best wishes
yours , Miaozhen Wang
Reviewer 2 Report
Good literature review
Author Response
Thank you very much for your attention and valuable and helpful comments.
Best regards
Sincerely yours